# Clinical Factors Predicting Multiple Endocrine Neoplasia Type 1 and Type 4 in Patients with Neuroendocrine Tumors

**DOI:** 10.3390/genes14091782

**Published:** 2023-09-10

**Authors:** Antongiulio Faggiano, Beatrice Fazzalari, Nevena Mikovic, Flaminia Russo, Virginia Zamponi, Rossella Mazzilli, Vito Guarnieri, Maria Piane, Vincenzo Visco, Simona Petrucci

**Affiliations:** 1Endocrinology Unit, Sant’Andrea Hospital, ENETS Center of Excellence, 00189 Rome, Italy; beatrice.fazzalari01@gmail.com (B.F.); nev.mikovic@gmail.com (N.M.); flaminia.russo@gmail.com (F.R.); virginia22.zamponi@gmail.com (V.Z.); rossella.mazzilli@uniroma1.it (R.M.); 2Department of Clinical and Molecular Medicine, Sapienza University of Rome, 00189 Rome, Italy; maria.piane@uniroma1.it (M.P.); vincenzo.visco1@uniroma1.it (V.V.); simona.petrucci@uniroma1.it (S.P.); 3Division of Medical Genetics, Fondazione IRCCS Casa Sollievo della Sofferenza, 71013 Foggia, Italy; v.guarnieri@operapadrepio.it; 4UOD Medical Genetics and Advanced Cell Diagnostics, Sant’Andrea Hospital, 00189 Rome, Italy

**Keywords:** multiple endocrine neoplasia type 1, multiple endocrine neoplasia type 4, neuroendocrine tumors, predictive factors, genetic test, phenocopies

## Abstract

The aim of this study is to evaluate the predictive role of specific clinical factors for the diagnosis of Multiple Endocrine Neoplasia type-1 (MEN1) and type-4 (MEN4) in patients with an initial diagnosis of gastrointestinal, bronchial, or thymic neuroendocrine tumor (NET). Methods: Patients referred to the NET Unit between June 2021 and December 2022 with a diagnosis of NET and at least one clinical criterion of suspicion for MEN1 and MEN4 underwent molecular analysis of the *MEN1* and *CDKN1B* genes. Phenotypic criteria were: (1) age ≤ 40 years; (2) NET multifocality; (3) MEN1/4-associated manifestations other than NETs; and (4) endocrine syndrome related to NETs or pituitary/adrenal tumors. Results: A total of 22 patients were studied. In 18 patients (81.8%), the first-level genetic test was negative (Group A), while four patients (25%) were positive for *MEN1* (Group B). No patient was positive for *MEN4*. In Group A, 10 cases had only one clinical criterion, and three patients met three criteria. In Group B, three patients had three criteria, and one met all criteria. Conclusion: These preliminary data show that a diagnosis of NET in patients with a negative family history is suggestive of MEN1 in the presence of ≥three positive phenotypic criteria, including early age, multifocality, multiple MEN-associated manifestations, and endocrine syndromes. This indication may allow optimization of the diagnosis of MEN in patients with NET.

## 1. Introduction

Multiple endocrine neoplasia (MEN) represents a group of genetic syndromes characterized by the occurrence of tumors within two or more endocrine glands.

Among MEN, the four most frequent nosological forms have been identified. MEN type 1 (MEN1, OMIM #131100) is an autosomal dominant disorder due to the mutation of the *MEN1* gene (NM_130799) responsible for producing a faulty or truncated version of the menin protein, which is unable to function as a tumor suppressor. Like MEN1, MEN type 2 (MEN2, previously MEN2A, OMIM #171400) and MEN type 3 (MEN3, previously MEN2B, OMIM #162300) are both autosomal dominant disorders due to *RET* gene (NM_020975) mutations, a gene responsible for producing a protein that is involved in the regulation of cell growth and division in several endocrine glands. Finally, MEN type 4 (MEN4, OMIM #610755) is a rare genetic disorder due to mutations in the cyclin-dependent kinase inhibitor (*CDKN1B*, NM_004064), which is located in chromosome 12p13 and encodes a protein called p27. The p27 protein plays a role in regulating cell division and preventing the uncontrolled growth of cells. Mutations in the *CDKN1B* gene can lead to a dysfunctional p27 protein [1].

In MEN1 and MEN4, genetic mutations and clinical manifestations are not directly related. For this reason, the genetic mutations are not found in all individuals with clinical manifestations of MEN, and not all individuals with the genetic mutations develop MEN [1].

The typical manifestations of MEN1 are (a) parathyroid hyperplasia/adenomatosis (>95%), (b) duodenal-pancreatic NETs (about 80%), and (c) pituitary adenoma (about 30–50%). Adrenal cortex tumors are less frequent and typical (25%), while lung, thymic, and gastric NETs occur in less than 10% of MEN1 patients [2]. MEN1-related NETs are slowly proliferating into G1-G2 malignant neoplasms. A somatic *MEN1* mutation is also found in the sporadic counterparts of MEN1 tumors: parathyroid adenoma (21%), gastrinoma (33%), insulinoma (17%), and bronchial carcinoid (36%). For each of them, the most frequently altered gene resulted to be *MEN1* [3]. MEN4 is the most recently identified type of MEN syndrome and shares a similar phenotype spectrum to MEN1. The most common presentation in MEN4 syndrome is primary hyperparathyroidism due to parathyroid tumors, followed by pituitary adenomas (functional and nonfunctional) and duodenal-pancreatic NETs [4]. A prompt and accurate diagnosis of MEN1 or MEN4 is critical for improving disease outcomes. This allows for early identification of clinical features, enabling timely treatment aimed at reducing morbidity and increasing survival. An appropriate management of MEN1 patients can be hampered by two challenges that currently hinder a proper diagnosis: diagnostic delay and phenocopies. The first one can be due to difficulties in detecting the proband, and to the delay in starting genetic screening in the affected family members [5,6,7,8,9]. To date, the lag time from the diagnosis of MEN1 in the proband to genetic analysis in the relatives is estimated to be 3.5 years [10].

Heterozygous damaging variants and variants of uncertain significance (VUS) can both be detected in MEN1 and MEN4 patients. VUS can occasionally be found during a genetic test for a specific condition, although it is not necessarily associated with harmful conditions.

The second challenge relates to “phenocopies”. Phenocopy is a term used in genetics to describe a condition or trait that mimics a genetic disorder but is not caused by a genetic mutation. Phenocopies can occur for several reasons, including environmental factors, overlapping syndromes due to different genetic mutations, or other non-genetic causes. Phenocopies might involve up to 5–25% of the MEN1 clinically diagnosed patients with no mutations in *MEN1* gene. At present, the clinical diagnosis of MEN1 requires the combined occurrence of at least two of the three typical manifestations. Patients that meet the clinically MEN1 criteria then undergo intensive follow-up. However, the validity of this criterion has been recently questioned. The simultaneous occurrence of primary hyperparathyroidism and pituitary adenoma is the most common MEN1 phenotype in patients with no mutations in *MEN1* gene but the same clinical manifestations have been also described in patients with damaging variants in *CDKN1B*, the causative gene of the recently described MEN4 syndrome [10].

In this study, we aim to focus on the patient with NET as the starting point for the diagnosis of MEN. The purpose of the study is to investigate, in this setting, the predictive phenotypic criteria for MEN1 and MEN4.

## 2. Materials and Methods

### 2.1. Patients

In this prospective observational study, all patients referred to the NET Unit of the Sant’Andrea Hospital of Rome from June 2021 to December 2022 with a diagnosis of duodenal-pancreatic (DP), gastric, bronchial, or thymic NET and at least one phenotypic criterion of suspicion for MEN1 or MEN4 underwent *MEN1* and *CDKN1B* molecular analysis. The clinical inclusion criteria required the presence of at least one among: (1) males and females aged ≤40 years; (2) NET multifocality; (3) MEN1 or MEN4-associated manifestations other than NETs, including primary hyperparathyroidism, pituitary adenoma, adrenal tumor/hyperplasia, meningioma, skin lesions (angiofibromas, lipomas, collagenomas, mucosal neuromas); (4) endocrine syndromes related to NETs as well as pituitary and adrenal tumors (Table 1). The study was conducted in accordance with the Declaration of Helsinki, and the protocol was approved by the Ethics Committee of Fondazione IRCCS Casa Sollievo della Sofferenza (approval identification number: prot N 13/CE-2021).

### 2.2. Methods

Genomic DNA samples from each proband were extracted from peripheral blood lymphocytes according to standard protocols. To search for nucleotide substitutions and small deletions/deletions of a few base pairs, molecular analysis of all coding exons and exon-intron boundaries of *MEN1* and *CDKN1B* was performed by next-generation sequencing (NGS) with the MiSeq Desktop Sequencer Platform (Illumina, San Diego, CA, USA), using a multigene panel that also included other genes responsible for disorders of osteocalcic metabolism (*RET*, *CDC73*, *AIRE*, *AP2S1*, *CASR*, *CYP24A1*, *GATA3*, *GCM2*, *GNA11*, *PTH*, *TBCE,* and *TBX1*) (first-level genetic test). Possible exonic rearrangements in the *MEN1* and *CDKN1B* genes were investigated with a multiple ligation-dependent probe amplification (MLPA) assay (SALSA MLPA Probemix P244 *AIP-MEN1-CDKN1B*, MRC-Holland, Amsterdam, the Netherlands) in probands with early onset or strong clinical suggestion of hereditary MEN1 (second-level genetic test).

In those patients with personal and family histories suggestive of other inherited cancer syndromes (such as Von Hipper Lindau syndrome, hereditary breast and ovarian cancer syndrome, familial melanoma, or inherited colorectal cancer syndromes), additional molecular analysis of related genes (*VHL*, *BRCA1*, *BRCA2*, *APC*, *ATM*, *BARD1*, *BRIP1*, *CDH1*, *CDK4*, *CDKN2A*, *CHEK2*, *EPCAM*, *MLH1*, *MRE11*, *MSH2*, *MSH6*, *MUTYH*, *NBN*, *PALB2*, *PMS2*, *PTEN*, *RAD50*, *RAD51C*, *RAD51D*, *RECQL1*, *SMAD4*, *STK11*, and *TP53*) by NGS (Ion PGM TM platform, Thermo Fisher Scientific, Waltham, MA, USA) and, when required, MLPA assays, have been considered.

The detected variants were classified according to the American College of Medical Genetics and Genomics (ACMG) criteria [11]. Damaging variants and variants of uncertain significance were confirmed by Sanger sequencing.

### 2.3. Statistical Analysis

The statistical analyses were performed using SPSS for Windows (version 20.0, SPSS, Inc., Chicago, IL, USA). The categorical variables were expressed as frequencies and percentage values. The comparison between the categorical variables has been assessed by the chi-square test and Fisher’s exact test for homogeneity. A *p* value of less than 0.05 was considered significant.

## 3. Results

A total of 27 consecutive patients were evaluated; of these, 22 patients met the inclusion criteria. After dedicated genetic counseling, molecular analysis was proposed to selected participants, and written informed consent was obtained from all of them. In 18/22 (81.8%), the first-level genetic test was negative (group A), while four (18.2%) had heterozygous variants in *MEN1* (group B): three had a damaging nucleotide substitution while the remaining carried a VUS (Table 2). In total, MLPA assays were performed in four patients, which were all negative. No patients had a damaging alteration or VUS in the *CDKN1B* gene.

Among the patients from group B, three carried a missense variant, while one had a splicing mutation (Table 2). According to the ACMG classification, two variants, c.658T > C, p.(Trp220Arg) and c.784-9G > A, p.?, were classified as pathogenic; the c.112 T > C, p.(Ser38Pro) substitution as likely pathogenic and the remaining, the c.563C > T, p.(Pro188Leu), was a variant of uncertain significance.

In group A, ten cases had one out of four predictive phenotypic criteria, five cases had two out of four phenotypic criteria, and three had three out of four criteria. In group B, three patients had three of the four criteria suggestive of MEN1, and one had all of the criteria (Table 3(A,B)).

In particular, among the 22 patients included in the study: (a) ten patients met the age criterion, with a first diagnosis of NET made before the age of 40; (b) nine patients complied with the criterion of multifocality of the NET at the time of the diagnosis; (c) 13 patients presented other MEN1 typical manifestations, including primary hyperparathyroidism (five cases), pituitary adenoma (four cases), adrenal adenoma/hyperplasia (three cases), meningioma (three cases), while the cutaneous lesions were the most represented with a total of 7 cases; (d) nine patients had endocrine syndromes: of these, three patients had insulinoma, two had Cushing’s syndrome, two had prolactinoma, one had Conn syndrome, one had carcinoid syndrome. In one patient, prolactinoma and Zollinger-Ellison syndrome coexisted (Table 3(A,B)).

The most frequent NET site in this series was the pancreas in 16 (72.7%), bronchopulmonary in six (27.3%), duodenum in two (10.1%), and stomach in one case (4.5%); three patients had both bronchial and pancreatic NET (Table 3(A,B)). The pancreas was the only tumor site predictive of MEN1 in patients with the first diagnosis of NET. The coexistence of pancreatic and bronchial NET did not specifically correlate with MEN syndrome. The association of pancreatic NET with primary hyperparathyroidism and/or pituitary adenoma and pituitary- or NET-related endocrine syndrome was predictive of MEN1, while the association with adrenal tumors, meningiomas, and skin lesions was not predictive. Young age and NET multifocality were less impactful. However, when both were present, together with another MEN manifestation or an endocrine syndrome, they were predictive of a positive genetic test. Family histories of NETs as well as other cancers were rare and not different between groups (Table 3(A,B)).

Other cancers were prevalent in non-mutated *MEN1* patients, while other manifestations were unspecific and not associated with any group.

By comparing the clinical features of probands with NETs with and without variants in *MEN1*, pancreatic NET was overrepresented in *MEN1* positive patients (4/4, 100%) but recurred also in *MEN1* negative cases (12/18, 66.7%) (*p* > 0.05) (Table 4). None of the four clinical factors considered was significantly prevalent in mutated vs. non-mutated subjects (*p* > 0.05). An early age of onset occurred in 75% of *MEN1* carriers and in 38.9% of *MEN1*-negative ones. No significant differences emerged comparing the occurrence of multifocal NETs and endocrine syndromes among the two groups. A trend was observed for the association of other MEN1-related manifestations and a positive molecular analysis (100% in mutated vs. 50% in non-mutated subjects, respectively), without any statistically significant difference (*p* > 0.05) (Table 4).

## 4. Discussion

MEN1 syndrome is an inherited disorder clinically characterized by the simultaneous presence of tumors in endocrine organs, including the pituitary gland, the parathyroid gland, and pancreatic islets. Specifically, parathyroid tumors are the most common feature of MEN1, occurring in about 95% of patients; neuroendocrine tumors of the duodenum-pancreas (i.e., nonfunctioning tumors, gastrinomas, insulinomas, pancreatic polypeptidomas, glucagonomas, and vasoactive intestinal polypeptidomas) occur in about 80% of patients. Furthermore, anterior pituitary adenomas (i.e., prolactinomas, somatotrophinomas, corticotrophinomas, or non-functioning adenomas) occur in about 30–50% of cases. Patients may also develop adrenal cortical tumors, carcinoid tumors, facial angiofibromas, collagenomas, and lipomas [12]. The estimated prevalence of MEN1 is low, ranging from 0.01 to 0.001% [13]. This is an autosomal dominant disorder with no gender prevalence and a variable age of onset (from 5 to 81 years).

The Clinical Practice Guidelines suggested that germline *MEN1* molecular analysis should to clinically diagnosed MEN1 probands and their first-degree relatives, including both asymptomatic subjects and those who have clinical manifestations of MEN1 [6,14].

In about 10% of patients with a MEN1-like phenotype in whom no *MEN1* mutation could be detected, other genes have been investigated. In some patients (up to 3%) with clinical diagnosis of MEN1 and with no damaging variants in *MEN1*, mutations in *CDKN1B* have been found. The *CDKN1B*-related clinical syndrome has been named MEN4 [15].

There are no definitive information about the exact incidence and prevalence of MEN4 because only very few cases of MEN4 have been described to date, and probably many are still undiagnosed. However, it is much lower than MEN1. In patients with MEN1-related neoplasia, a MEN4 is likely to be around 3% [1].

NETs are a rare type of cancer that originates from neuroendocrine cells, which are dispersed throughout the body. For this reason, these tumors can occur in various organs of the body, such as the pancreas, lungs, gastrointestinal tract, and other parts of the body.

The overall incidence of NETs is much higher than MEN, being reported at around 7 cases per 100,000 per year [16], while the prevalence is 0.048%. Of consequence, only a small subgroup of NET patients will finally be affected by MEN1 and even less by MEN4. The association between MEN and NET is particularly strong with pancreatic NETs. Approximately 10–20% of all pancreatic NETs are associated with MEN1, and up to 90% of patients with MEN1 will develop pancreatic NETs during their lifetime. In addition to pancreatic NETs, individuals with MEN1 may also develop NETs in other parts of the body, including the thymus, lungs, and gastrointestinal tract [13].

A large French series highlighted a shorter delay to achieve the diagnosis of MEN1 when the first manifestation of the syndrome is a NET, as compared to hyperparathyroidism or pituitary adenomas [5]. This finding suggests that patients with NET are more frequently investigated for MEN syndromes than those with parathyroid and pituitary tumors. In the current study, we investigated which clinical features are predictive for MEN1 or MEN4 syndrome in order to improve the diagnostic work-up by correctly identifying the candidates for genetic investigations and by avoiding unnecessary genetic analyses with relative costs and stress for the patient.

All the *MEN1* gene variants identified in patients from group B were previously described. The c.784-9G > A splicing variant, which creates a novel splice acceptor site resulting in a frameshift and a subsequence premature stop codon [17,18], has been detected in unrelated MEN1 patients and families [17,18,19,20,21]. Also, the damaging missense variants c.112T > C and c.658T > C, extremely rare in general population databases (GnomAD Exomes and Genomes) and predicted to be disruptive to the protein, have been described in MEN1 cases [22,23]. Conversely, the c.563C > T substitution, although observed in cases with non-familial primary hyperparathyroidism and pituitary adenoma [24,25,26,27], is not so rare in general population (rs199706698, gnomAD 0.01%), and the predictions on its consequences on protein structure and/or function are still conflicting. However, we detected it in a proband with a MEN1 diagnosis from the age of thirteen (pancreatic NET, pituitary microadenoma, and collagenomas on her back and her head) and in his father, with adult-onset pancreatic NET and adrenal adenoma. To date, the pathogenicity of this variant is controversial. However, no other deleterious alterations in *MEN1* and *CDKN1B* have been identified in this family. Even if we cannot exclude the presence of other variants undetectable with available techniques, the co-segregation of c.563C > T with the phenotype may suggest a possible contribution of this variant to the occurrence of the disease.

The main finding of the study confirms that pancreatic NETs and no other NET sites represent a starting point for a possible MEN1 or MEN4 syndrome. The incidence of pancreatic NET is 1.5 cases per 100,000, and the prevalence is 0.008% [26]. In a large Italian series of NETs, MEN1 involved 7% of patients with gastroenteropancreatic or thoracic NETs [27]. This rate was higher in pancreatic NETs and, in particular, in gastrinomas, where it has been estimated to affect 16–38% of cases [6].

If the genetic investigation for MEN1 and MEN4 starts with a diagnosis of pancreatic NET, at least three other conditions need to be encountered to suggest the diagnosis of an inherited syndrome, including age, multifocality, endocrine syndrome, and more importantly, other MEN1 manifestations. When two or just one of these criteria occur, a MEN1 or MEN4 has never been diagnosed by molecular analysis. These cases have to be assumed to be sporadic NET and have not to undergo MEN1 and MEN4 genetic tests. Also, patients with three of the four criteria are frequently non-mutated for MEN1 or MEN4 syndrome, although the occurrence of pituitary adenoma or primary hyperparathyroidism increases the chance of a positive genetic test. On the contrary, adrenal and skin lesions, as well as meningiomas, are less specific for MEN1 and MEN4. Only one patient accomplished all the criteria, resulting in a positive result for *MEN1*. To summarize, patients with ≥3 phenotypic criteria should undergo *MEN1* and *CDKN1B* molecular analysis.

Since nucleotide substitutions and small nucleotide deletions/insertions are the most frequently encountered alterations, sequencing of all coding regions and intron-exon boundaries should be the first-level genetic test. If sequencing is not informative, a second-level genetic test based on quantitative analysis, such as the MLPA assay, should be performed to detect rarer exonic or multiexonic deletions (about 2.5% of the identified *MEN1* pathogenic alterations) [25]. As no direct genotype-phenotype correlations have been identified for *MEN1* and *CDKN1B* alterations, the second-level genetic test should be considered in all patients who meet ≥3 phenotypic criteria for MEN1 [9,28,29]. This would result in a >50% rate of positive genetic tests.

It is not surprising that no case of MEN4 has been found in this small series. In fact, MEN4 is a very rare syndrome, and no clinical predictive factors have been identified. It is usually searched for in MEN1 patients with no alterations at the *MEN1* molecular analysis. In MEN4, primary hyperparathyroidism has been reported in up to 80–90% of cases and presents at an older age as compared to MEN1, with a female predominance [6,7]. The second most common presentation is pituitary NET. As in MEN1, acromegaly is expected to be found in 10% of MEN4 cases, whereas Cushing disease occurs in 5% of patients [8]. Unlike MEN1, the frequency of gastrinomas and nonfunctional pancreatic NETs, estimated to be in 25% of MEN4 cases, is significantly lower than MEN1. So far, no cases of insulinoma, VIPoma, glucagonoma, or ectopic ACTH-secreting NETs in MEN4 have been reported [9]. On the other hand, there have been reports of adrenal tumors, testicular cancer, cervical carcinoma, papillary thyroid cancer, colon cancer, carcinoids, and meningiomas in MEN4 [8,9].

In line with literature data, our study confirms that in patients with pancreatic NETs, the co-occurrence of other MEN1 manifestations, such as primary hyperparathyroidism and pituitary adenoma, is a strong predictive feature for damaging variants in *MEN1* gene. However, some patients with a clinical phenotype highly suggestive of MEN1 or MEN4 remain negative for all the genetic investigations. Other NET primaries (i.e., lung, stomach, etc.) and other MEN1-related manifestations, such as adrenal lesions, skin lesions, and meningiomas, are not specific for these syndromes, as are young age, NET multifocality, or both, when not associated with other MEN1 manifestations. The early onset of the disease has been reported to be closely related to *MEN1*-mutated carriers [30]. However, in our cohort, statistically significant differences in age of onset among *MEN1* carriers and *MEN1* non-carriers did not emerge. Furthermore, in the real world, many patients with sporadic NET are diagnosed before 40 years of age, and sometimes before 20 years. This can be due to the increased interest and improved diagnostic tools that allow an early diagnosis of NET in many patients. On the other hand, some MEN1 patients without a family history are diagnosed after 40 years of age. Therefore, the age of NET diagnosis cannot be considered a criterium to start a genetic work-up in patients with NET if no other predictive features occur.

In order to perform a diagnosis of MEN, an accurate anamnesis and clinical examination are needed. Based on clinical suspicion, the patients should undergo genetic counseling and subsequent genetic testing. Obtaining a VUS result on genetic testing represents a real challenge for the clinician. In fact, to date, there are not enough data on the meaning of VUS or on follow-up to offer the patient [6].

The main limitation of the study is the small sample size of the cohort. Furthermore, molecular analysis able to investigate deep intronic variants that may alter the splicing process, variants in regions that may alter *MEN1* or *CDKN1B* expression, and epigenetic alterations are not available.

## 5. Conclusions

The main interest of the study arises from the paucity of studies exploring the correlation between genotype and phenotype in MEN1 and MEN4 syndromes. Another original aspect is to focus on patients with NETs as the starting point for the genetic work-up. In patients with NET, especially if of pancreatic origin and associated with other factors such as other MEN1-related manifestations, age, multifocality, and associated endocrine syndromes, genetic screening is useful to achieve an early diagnosis of MEN, set up related treatment and follow-up, and start the genetic screening in first-degree relatives. Further multi-center studies based on a larger population are needed to better clarify the role of phenotypic criteria in the diagnostic workup of MEN 1 and MEN4.

## Figures and Tables

**Table 1 genes-14-01782-t001:** Phenotypic criteria for MEN1 and MEN4.

Criteria	Cut Off
**Age**	≤40 years
**NET multifocality**	≥2 NET lesions within the primary site
**Other manifestations other than NET**	Primary hyperparathyroidism Pituitary adenoma AdrYes, we confirmenal tumour/hyperplasia Meningioma Skin lesions (angiofibromas, lipomas, collagenomas, mucosal neuromas)
**Endocrine syndrome related to NETs**	Pituitary-and adrenal-related

NET—neuroendocrine tumour.

**Table 2 genes-14-01782-t002:** Patients with a positive *MEN1* mutation analysis.

dbSNP	cDNA (HGVS)	Protein	Type of Variant	ACMG Classification
rs1341908127	c.112T > C	p.(Ser38Pro)	missense	LP
rs794728625	c.784-9G > A	p.?	splicing	P
rs1085307971	c.658T > C	p.(Trp220Arg)	missense	P
rs199706698	c.563C > T	p.(Pro188Leu)	missense	VUS

*MEN1* (NM_001370259.2) variants of Group B patients. LP—likely pathogenic; P—pathogenic; VUS—variant of uncertain significance; ACMG—American College of Medical Genetics and Genomics.

**Table 3 genes-14-01782-t003:** (A) Predictive phenotypic criteria in non-mutated NET patients. (B) Predictive phenotypic criteria in *MEN1*-mutated NET patients.

**(A)**									
**Pt**	**NET**	**Age < 40 Years**	**NET Multifocality**	**Other than NET MEN1 or MEN4-Related Manifestations**	**Endocrine Syndromes**	**Family History of NET**	**Family History of Other Cancers**	**Other Cancers**	**Other Clinical Manifestations**
DF	Pancreatic G1	No	No	No	Insulinoma	No	yes		Chronic autoimmune thyroiditis
DD	Gastric G1	Yes	No	No	No	No	yes		Atrophic gastritis, oesophagitis, hiatal hernia, nodular thyroid disease
MA	Pancreatic G1	No	No	PHPT	No	No	Yes		Multinodular thyroid disease
CF	Bronchial	No	Yes	PHPT, meningioma, mucosal neuroma	Carcinoid syndrome	Yes	Yes	Chronic myeloid leukemia	Thyroid nodule, liver and kidney cysts, GERD, aortic and mitralic regurgitation
DDA	Duodenal G2	No	Yes	No	No	No	Yes	Prostate adenocarcinoma	Nodular thyroid disease, hepatic and splenic cysts, gallbladder lithiasis, hypertension, aortic aneurysm, OSAS
XA	Pancreatic G2	Yes	No	Adrenal adenoma	No	No	No		Hypothyroidism, ovarian and renal cysts, diabetes type 2
SC	Pancreatic G2	Yes	No	No	No	No	No		
CMR	Pancreatic G1	No	Yes	Meningioma, angiofibromas	No	No	Yes		Pancreatic and liver cysts, diabetes type 2, hypertension, nodular thyroid disease, talassemia
MD	Pancreatic G1	Yes	No	ACTH-secreting PA	Cushing syndrome	No	No		
CL	Pancreatic G2	Yes	No	No	No	No	No	Melanoma	Bronchial asthma, gastropathy, hypertension
MI	Bronchial/Tumourlets	No	Yes	No	No	No	No	Chromophobe cell renal carcinoma	Nodular thyroid disease, breast fibroadenomas, uterine fibromatosis, pancreatic and kidney cysts
FL	(1) Bronchial (2) Pancreatic G1	No	No	Adrenal hyperplasia	No	No	Yes		Congenital glaucoma, hypertension, facial paresis, liver and kidney cysts
BS	Pancreatic G2	No	No	Meningioma, lipomas and collagenoma	No	No	Yes		
RMS	Bronchial	No	Yes	No	No	No	Yes		Pancreatic cyst, hepatic cystic-dysplastic area
ZO	Duodenal G2	Yes	No	Adrenal adenoma	Conn syndrome	No	No		Kidney cyst
AC	Pancreatic G1	Yes	No	No	Insulinoma	No	No		Autoimmune thyroid disease
MS	Pancreatic G2	No	Yes	No	Insulinoma	No	No		Nodular thyroid disease
BA	(1) Pancreatic G1 (2) Bronchial	No	No	Lipomas	Cushing syndrome	No	No		Hypertension, diabetes type 2, hypothyroidism, rectal prolapse
**(B)**									
**Pts**	**NET**	**Age < 40 Years**	**NET Multifocality**	**Other than NET MEN1 or MEN4-Related Manifestations**	**Endocrine Syndromes**	**Family History of NET**	**Family History of Other Cancers**	**Other Cancers**	**Other Clinical Manifestations**
MM	Pancreatic G1	Yes	No	PHPT, lipomas, collagenomas, angiofibromas	Glucagonoma	Yes	No		Thyroid nodules
GN	Pancreatic G1 Bronchial	No	Yes	PHPT, PA	Prolactinoma	No	No		Diabetes type 2, GERD
MM	Pancreatic G2	Yes	Yes	PHPT, PA, lipomas, angiofibromas	Prolactinoma, ZES	No	No		Liver cyst, Breast adenoma
NL	Pancreatic G1	Yes	Yes	PA, collagenomas	no	No	No		

NET—neuroendocrine tumour; PHPT–primary hyperparathyroidism; PA—pituitary adenoma; ZES: Zollinger-Ellison syndrome.

**Table 4 genes-14-01782-t004:** Comparison of clinical features between mutated and non-mutated *MEN1* patients.

	Mutated	Non-Mutated	*p* Value
PanNET	4/4 (100%)	12/18 (66.7%)	>0.05
Family history of NET	1/4 (25%)	1/18 (5.6%)	>0.05
Age ≤ 40 yrs	3/4 (75%)	7/18 (38.9%)	>0.05
NET multifocality	3/4 (75%)	6/18 (33.3%)	>0.05
MEN1 and MEN4-associated manifestations other than NET	4/4 (100%)	9/18 (50%)	>0.05
Endocrine syndrome	3/4 (75%)	7/18 (38.9%)	>0.05

PanNET—pancreatic neuroendocrine tumor.

## Data Availability

The data presented in this study are available on request from the corresponding author. The data are not publicly available due to privacy restrictions.

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
