# Peer review of "Clinical Factors Predicting Multiple Endocrine Neoplasia Type 1 and Type 4 in Patients with Neuroendocrine Tumors"

_genes, 2023, doi:10.3390/genes14091782_

Round 1

Reviewer 1 Report

The current manuscript is entitled: Clinical factors predicting multiple endocrine neoplasia type 1  and type 4 in Patients with neuroendocrine tumors by Faggiano et al and colleagues investigated the predictive role of specific clinical factors for the 17 diagnoses of Multiple Endocrine Neoplasia type-1 (MEN1) and type-4 (MEN4), in patients with in- 18 initial diagnoses of gastrointestinal, bronchial, thymic neuroendocrine tumor (NET). The authors conducted a study on 22 patients and shows that, out of these, 18 patients (81.8%) tested negative for the first 24-level genetic test (Group A), while four patients (25%) tested positive for MEN1 (Group 25 B). Furthermore, the authors found that none of the patients tested positive for MEN4. The authors show that a diagnosis of NET in patients with a negative family 28 history is suggestive of MEN1 in presence of ≥ three positive phenotypic criteria, including early 29 age, multifocality, multiple MEN-associated manifestations, endocrine syndromes. The authors have done a great job writing the manuscript in good English. This manuscript has great clinical impact; therefore, I recommend this article for publication in genes.

English is good

Author Response

We thank the Reviewer for the positive evaluation of our work.

Reviewer 2 Report

This study used 22 patients to assess the association of MEN1 and MEN4 with NET and it was found that MEN1 was positive in 25% of the patients but all negative for MEN4. It appears that MEN1 has better predictive value in NET diagnosis. The manuscript is clearly written. It would be sound if the study population were greater.

A few minor errors were found. The authors are suggested to go through the paper carefully.

Author Response

This study used 22 patients to assess the association of MEN1 and MEN4 with NET and it was found that MEN1 was positive in 25% of the patients but all negative for MEN4. It appears that MEN1 has better predictive value in NET diagnosis. The manuscript is clearly written. It would be sound if the study population were greater.

R: We thank the Reviewer very much for the comment. We better specify the main limitation of the study as follows: “The main limitation of the study is the small sample size of the cohort. Furthermore,  molecular analysis able to investigate deep intronic variants that may alter splicing process, variants in regions that may alter MEN1 or CDKN1B expression and epigenetic alterations are not available.”.

A few minor errors were found. The authors are suggested to go through the paper carefully.

R: The manuscript was revised and language check was done.